https://doi.org/10.1038/s43856-022-00217-1　　OPEN
# Whole-muscle fat analysis identifies distal muscle end as disease initiation site in facioscapulohumeral muscular dystrophy

Linda Heskamp [1✉], Augustin Ogier[2], David Bendahan[2] & Arend Heerschap [1✉]

## Abstract

**Background** Facioscapulohumeral dystrophy (FSHD) is a major muscular dystrophy characterized by asymmetric fatty replacement of muscles. We aimed to determine the initiation site and progression profile of the disease in lower extremity muscles of FSHD patients by assessing fat infiltration along their full proximo-distal axis using quantitative MRI.

**Methods** Nine patients underwent MRI of lower extremities to assess end-to-end muscle fat fractions (FFs) and inflammatory lesions. Seven patients underwent the same MRI ~3.5 years later. Individual muscles (n = 396) were semi-automatically segmented to calculate average FFs over all slices covering whole muscles. To assess disease progression we determined FF changes in 5 adjacent muscle segments.

**Results** We provide evidence that fat replacement commonly starts at the distal end of affected muscles where the highest FFs occur (p < 0.001). It progresses in a wave-like manner in the proximal direction at an increasing rate with the highest value (4.9 ± 2.7%/year) for muscles with baseline FFs of 30–40%. Thereafter it proceeds at a slower pace towards the proximal muscle end. In early phases of disease, inflammatory lesions preferentially occur at the distal muscle end. Compared with whole-muscle analysis, the common FF assessments using only few MR slices centrally placed in muscles are significantly biased (~50% in progression rate).

**Conclusions** These findings identify the distal end of leg muscles as a prime location for disease initiation in FSHD and demonstrate a wave-like progression towards the proximal end, consistent with proposed disease mechanisms. End-to-end whole-muscle fat assessment is essential to properly diagnose FSHD and its progression.

### Plain language summary

Infiltration of fat in muscle is a feature of muscular diseases. One example is facioscapulohumeral muscular dystrophy. Here, we investigated where fat infiltration starts and how it progresses in leg muscles of patients with this disorder. We used magnetic resonance imaging to visualise the fat content of all the leg muscles. This showed that in nearly all affected muscles, fat infiltration begins in the muscles' extreme lower end, which means that disease starts at this end. Subsequently, fat infiltration progresses as a wave towards the muscle's upper end. Our observations also suggest that assessing fat content in whole muscle, rather than the common approach of only assessing the middle part of muscles, measures fat infiltration more accurately. These findings are relevant to identify factors involved in disease onset, to develop and evaluate therapies, and in disease diagnosis.

[1] Department of Medical Imaging/Radiology, Radboud university medical center, Nijmegen, The Netherlands. [2] Aix Marseille Univ, CNRS, CRMBM, Marseille, France. ✉email: l.heskamp-2@umcutrecht.nl; Arend.heerschap@radboudumc.nl

Facioscapulohumeral dystrophy (FSHD) is the second most common type of muscular dystrophy worldwide. The disease is characterised by progressive skeletal muscle weakness, with the first signs often observed in the upper arm, shoulder, and face muscles, followed by the trunk and lower extremity muscles[1,2].

The genetic cause of FSHD is a contraction of the D4Z4 repeat in the subtelomeric region of chromosome 4q, which releases suppression of the *DUX4* (Double HomeoBox 4) gene. Subsequently, an unknown initiation process causes expression of the DUX4 transcription factor followed by a complex cascade of events that eventually leads to muscle cell death[3,4]. The search for new treatments for FSHD focuses on this genetic abnormality and the suppression of DUX4 or its targets[5,6]. However, information about disease initiation and progression are of paramount importance for the development and evaluation of successful treatments[7].

The progressive loss of muscle strength in FSHD is strongly related to the replacement of muscle tissue with fat and muscle cell atrophy[8–11]. MRI is an ideal tool for the investigation of these pathological processes as it can quantitatively assess the relative amount of fat in skeletal muscle tissue[12]. MR studies conducted on FSHD patients have revealed that some muscles are more susceptible to fat infiltration than others[8–10,13–15]. Furthermore, several of these studies hint towards a higher fat content in the distal direction of affected muscles, suggesting the presence of a linear gradient of increasing, proximo-distal fat content[10,13,16,17]. However, these MRI studies were performed on a limited number of transversal slices located at the muscle belly; therefore, it is unknown whether this proximo-distal intra-muscular fat gradient extends over the whole muscle or displays, for example, a U-shape with the highest fat infiltration at both muscle ends. The profile of this fat distribution obviously has consequences for hypotheses about disease initiation in the muscles of FSHD patients.

For these reasons, the first aim of our study was to perform a quantitative cross-sectional MR evaluation of fat infiltration covering the whole length of diseased muscles, instead of only a central section. It has been proposed that DUX4 molecules, generated only in a few myonuclei, can activate target genes in nearby nuclei and cause a transcriptional cascade of DUX4-mediated dysregulation that travels along the whole muscle[7,18,19]. Therefore, we hypothesised that fat infiltrates diseased muscles more as a wave instead of gradually increasing along a linear gradient over the length of the muscles. To test this hypothesis, our second aim was to evaluate the progression of fat infiltration using quantitative MRI along the whole length of the proximo-distal axis in muscles of FSHD patients over a period of about 3.5 years. This evaluation required the complete end-to-end analysis of the muscles using a large number of slices. To overcome the high workload of muscle delineation in all these slices, we used a semi-automatic segmentation tool[20].

We detect that in nearly all affected muscles, fat infiltration starts at the distal end and progresses as a wave towards the proximal end of the muscles. We also observe that the common image analyses of fat in only the middle part of muscles may inaccurately assess fat infiltration.

## Methods

**Study design and participants**. Nine FSHD patients were recruited for this study. Exclusion criteria were contraindications for MRI scanning, including claustrophobia, an implanted pacemaker, or being unable to lie supine for 60 min. After the patients completed their first MR exam, they were invited for a follow-up MRI 3.5 years later.

This study was conducted according to the principles of the Declaration of Helsinki (updated October 2013) and in accordance with the Medical Research Involving Human Subjects Act (WMO). The study was approved by the local medical ethics committee CMO Arnhem-Nijmegen (IRB approval number: NL66071.091.18) and all participants gave written informed consent.

**MRI acquisition**. All participants underwent an MRI examination of their lower extremity muscles using 3 T MR systems (Siemens, Erlangen, Germany) with a spine coil and phased array coils placed around the lower extremities. Axial 3D 2-pt Dixon images were acquired to quantify muscle fat infiltration. The lower extremity was covered from hip to ankle using 3 or 4 stacks. The most proximal stack was placed with the proximal slice at the head of the femur and each subsequent stack was placed distal to the former stack with a 45 mm overlap. Baseline MRI scans were performed on a Siemens Tim TRIO with the following settings: repetition time = 10 ms, echo times in-phase/out-phase = 2.45/3.68 ms, flip angle = 3°, field of view = 435 × 271 mm, in-plane resolution = 1.36 × 1.36 mm, slice thickness = 5 mm, number of slices = 72). Due to an upgrade of the MR system between the initial scan and follow-up, the follow-up MRI examinations were acquired on a Siemens Prisma with similar settings, except that the out-phase/in-phase echo times were 1.26 and 2.49 ms, respectively. Furthermore, inflammatory lesions were assessed with a Turbo-Inversion Recovery Magnitude (TIRM) sequence using the following settings at both baseline and follow-up: repetition time = 4140 ms, echo time = 41 ms, inversion time = 240 ms, flip angle = 150°, field of view = 435 × 271 mm, in-plane resolution = 1.36 × 1.36 mm, slice thickness = 5 mm, and slice gap = 5 mm[21]. The subsequent stacks were aligned in the feet-head direction, with no overlap.

**MRI analysis**. MR data were analysed using Matlab version 2018a (Mathworks, Natick, MA, USA). A fat fraction map was calculated from the Dixon fat and water images reconstructed by Siemens Syngo Via software (fat image/[fat image + water image]). The resulting values ranged from 0 to 100% fat. Because the MR system upgrade changed the Dixon reconstruction software, we used the Dixon images from four FSHD patients examined just before and after the upgrade to correct the baseline fat fraction maps. The relationship between the fat fraction before and after the upgrade fit a linear model (baseline fat fraction corrected = 1.1864 × baseline fat fraction – 2.7878; $r^2 = 0.994$).

Subsequently, we combined the separate stacks into a single dataset covering all lower extremity muscles from hip to ankle. Baseline and follow-up scans were aligned in the feet-to-head direction to ensure both had the same proximo-distal coverage. For every leg, 12 upper leg and 10 lower leg muscles were manually segmented on every fifth slice by an experienced observer (LH). These segmentations were used to automatically segment the remaining slices by propagation (Fig. 1)[20,22]. This semi-automatic segmentation was verified by the experienced observer and manually corrected where needed.

The segmentations of individual muscles were used to calculate the average fat fraction of the whole muscle, including all slices, both for baseline and follow-up. Subsequently, a composite fat fraction of the whole lower extremity of each participant was estimated by taking the weighted average of the fat fractions of the individual muscles, i.e. the fat fraction of each muscle was normalised by the volume of that muscle; thus, larger muscles have more weight than smaller muscles.

Similarly, weighted composite fat fractions were calculated for the upper leg, lower leg, and for several functional muscle groups. The change in a fat fraction over time was then calculated for individual muscles and for composite scores, and was expressed as

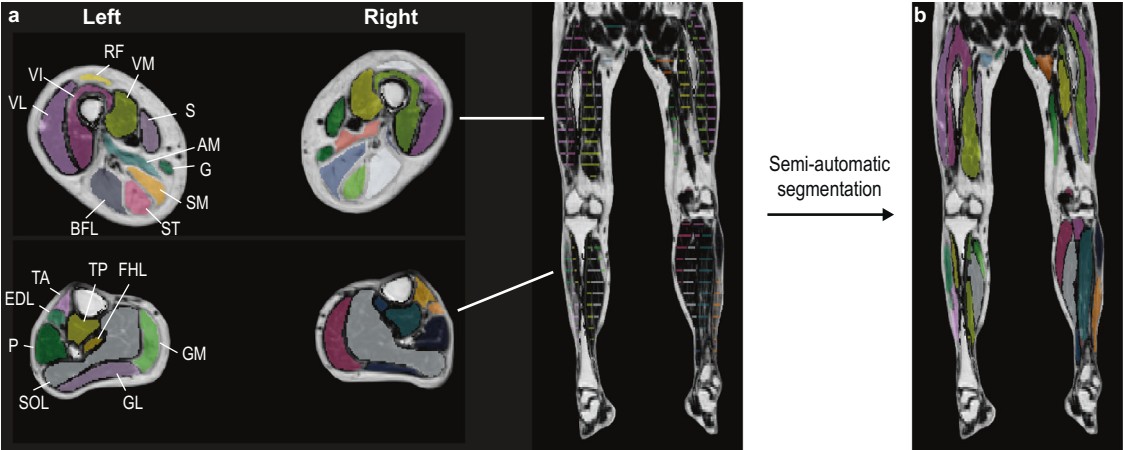

**Fig. 1 Overview of manual and semi-automatic segmentations of lower extremity muscles displayed on fat fraction maps.** Each muscle was manually segmented at every fifth measured slice from its distal to the proximal end (**a**). The slices were then propagated, as published by Ogier et al. (2017), to segment all remaining slices (**b**). As a result, all muscles were segmented on every slice (every 5 mm). The colours represent the different muscles. Upper leg muscles: AM adductor magnus, BFL biceps femoris long head, G gracilis, RF rectus femoris, S sartorius, SM semimembranosus, ST semitendinosus, VI vastus intermedius, VL vastus lateralis, VM vastus medialis. Lower leg muscles: EDL extensor digitorum longus, FHL flexor hallucis longus, GM gastrocnemius medialis, GL gastrocnemius lateralis, P peroneus, SOL soleus, TA tibialis anterior, TP tibialis posterior.

fat fraction change (follow-up fat fraction – baseline fat fraction) and fat fraction change per year (follow-up fat fraction – baseline fat fraction/follow-up time in years).

To evaluate fat infiltration along the proximo-distal axis, we also calculated the fat fraction per slice for each muscle. Because muscle length and, therefore, the number of segmented slices differed between muscles and participants, slice numbers were expressed as a percentage of the muscle length, with the most distal slice set at 0% and the most proximal slice set at 100%. Furthermore, we divided each muscle into five equally spaced proximo-distal segments, again relative to the muscle length, and calculated the average fat fraction per segment. For example, the most distal 1/5th of the muscle was set at 0–20% and the most proximal 1/5th of the muscle was set at 80–100% of its length.

The TIRM images were reviewed by an experienced musculoskeletal radiologist and scored for the presence of signal hyperintensity. To determine at which position along the muscle length the TIRM hyperintensity preferentially occurred, we visually scored the location of the TIRM-positive lesions for all TIRM-positive muscles with low or mild fat infiltration (<30% fat fraction). For this purpose, each muscle was evenly divided into three parts: distal, middle and proximal.

**Statistics**. Statistical analysis was performed using IBM SPSS Statistics (version 25, Chicago, IL, USA). We applied several linear mixed models to manage the nesting of muscles in patients (muscle data were level 1 and patient data were level 2). The covariance structure was set at variance components and the models were fitted with a restricted maximum likelihood estimation.

The proximo-distal fat infiltration at baseline was assessed with a linear mixed model, with segment location as a fixed covariate (i.e., fat fraction $= \beta_0 + \beta_1 \times$ segment) and intercept modelled as a random effect. This analysis was performed for all muscles and by categorising muscles based on the Mercuri scoring system into low fat-infiltrated muscles (<10% fat), mildly fat-infiltrated muscles (10–30% fat), moderately fat-infiltrated muscles (30–60% fat), and severely fat-infiltrated muscles (>60% fat)[23].

The longitudinal change in fat fraction at the participant level was evaluated with a two-sided Wilcoxon signed rank test. The effect of the baseline fat fraction on this longitudinal change was calculated by dividing the muscles into groups based on their baseline fat fraction (0–10%, 10–20%, 20–30%, 30–40%, 40–50%, 50–60%, 60–70%, 70–80%, and 80–90%). We then applied a linear mixed model to compare the change in a fat fraction in each group to muscles with a baseline fat fraction of 0–10%. The baseline fat fraction group number was used as a fixed factor and the intercept was modelled as a random effect. We corrected for multiple testing with Bonferroni (number of tests = 8).

In quantitative MR studies of FSHD patients, it is common to analyse only a limited number of transversal slices. However, if fat infiltration is not homogeneously distributed along the muscle, fat fractions can be under- or over-estimated. To estimate the size of this effect in FSHD, we compared the baseline fat fraction and change in fat fraction determined over the whole muscle to these values assessed over the five middle slices or over five slices equally spread along the muscle, i.e., at 10%, 30%, 50%, 70%, and 90% of the muscle length. To this end, we calculated for each muscle the error and absolute error in baseline fat fraction and change in fat fraction. We defined error as the fat fraction (or change in fat fraction) obtained from five slices minus the fat fraction (or change in fat fraction) obtained over the whole muscle, and the absolute error is the absolute value of this difference. The average error over all muscles and limits of agreement were obtained with Bland–Altman analysis. Furthermore, we tested whether the absolute error was different from 0 using a linear mixed model, with intercept modelled as a random effect and no added covariate or factor. For the linear mixed model results, the absolute error estimated by the model is reported.

Statistical significance was set at $p < 0.05$. The results of the linear mixed models are displayed as estimates (standard error [SE]), and all other data were presented as mean ± standard deviation (SD) unless stated otherwise.

**Reporting summary**. Further information on research design is available in the Nature Portfolio Reporting Summary linked to this article.

## Results

**Participants**. We included nine patients (seven male and two female) diagnosed with FSHD type 1 and with an average age of 61 years (range: 46–72 years). Seven of these patients (seven male) underwent a follow-up MRI scan 3 years and 8 months

**Table 1 Average baseline fat fraction and change in fat fraction observed in the facioscapulohumeral muscular dystrophy (FSHD) patients.**

| | Baseline fat fraction (%) (Mean ± SD; $n = 9$) | Change in fat fraction (Mean ± SD; $n = 7$) | Change in fat fraction per year (Mean ± SD; $n = 7$) |
|---|---|---|---|
| Lower extremity | 32.1 ± 16.2 | 4.7 ± 5.9 | 1.3 ± 1.6 |
| Upper legs | 36.0 ± 17.8 | 4.7 ± 6.6 | 1.3 ± 1.8 |
| Anterior compartments | 27.2 ± 21.0 | 2.8 ± 5.1 | 0.8 ± 1.4 |
| Posterior compartments | 57.7 ± 20.7 | 4.9 ± 7.6 | 1.3 ± 2.1 |
| Medial compartments | 39.7 ± 18.5 | 9.5 ± 10.6 | 2.5 ± 2.8 |
| Lower legs | 25.0 ± 13.6 | 4.1 ± 4.7 | 1.1 ± 1.3 |
| Anterior compartments | 27.6 ± 16.4 | 1.2 ± 2.0 | 0.3 ± 0.6 |
| Posterior compartments | 26.1 ± 16.5 | 4.9 ± 5.6 | 1.3 ± 1.5 |
| Lateral compartments | 19.1 ± 9.7 | 2.2 6.3 | 0.6 ± 1.7 |

Fat fractions were calculated for individual muscles and subsequently averaged to calculate weighted composite scores for the lower extremity, upper leg, lower leg, and the muscle groups, both left and right leg averaged. Standard deviations for weighted composite scores in individual patients are not presented, as some muscle groups consisted of only one or two muscles. Upper leg: anterior compartment (rectus femoris, vastii muscles and sartorius), posterior compartment (biceps femoris, semitendinosus and semimembranosus), medial compartment (adductor magnus, adductor longus and gracilis). Lower leg: anterior compartment (tibialis anterior and extensor digitorum longus), posterior compartment (gastrocnemius, soleus, tibialis posterior, flexor hallucis longus, flexor digitorum longus and popliteus), lateral compartment (peroneus). Baseline fat fraction and change in fat fraction values of individual participants can be found in the supplemental Table S1.

after the baseline scan (range: 3 years and 5 months to 3 years and 10 months).

The evaluation included a total of 386 muscles at baseline and 298 muscles at follow-up, whereby on average, 45 slices (range: 6–91 slices) were analysed per muscle. Ten lower leg muscles were excluded from further analysis because the Dixon fat-water reconstruction for the baseline MRIs resulted in a fat-water swap in the middle of one lower leg.

**Muscle fat fraction at baseline**. The average fat fraction at baseline over all muscles and patients was 32.1 ± 16.2% (mean ± SD), ranging from an average fat fraction over all muscles of 8.7% for the least affected patient to 52.4% for the most affected patient. In individual muscles, whole muscle fat fraction varied from 3.1 to 89.5%. The upper leg muscles showed the highest fat fractions (Table 1 and see Supplemental Table S1 for individual participant fat fraction values).

An evaluation of the fat infiltration within individual muscles revealed a declining distal-to-proximal fat content in muscles with a mild fat fraction (10–30%, $n = 102$) and a moderate fat fraction (30–60%, $n = 68$), as illustrated for the gastrocnemius medialis, vastus lateralis and biceps femoris long head (Fig. 2a–d). In nearly all these cases, the fat content along the whole muscle length appears to follow a (shifted) reversed sigmoid curve of which the slope steepness decreases at higher overall muscular fat contents. This distal-to-proximal curve of declining fat content also occurred in some muscles with an overall low-fat fraction (<10%, $n = 121$) and a high-fat fraction (>60%, $n = 95$), but in particular for the latter fat fraction group, most muscles showed no distinct lower fat content proximally.

To evaluate intra-muscular fat infiltration quantitatively, we determined the median fat content in five segments along the length of all muscles. For this evaluation, muscles were divided into four groups of increasing overall baseline muscular fat fraction (Fig. 2e–h). The median fat fraction for a given segment depended on its position along the proximo-distal axis. Including all investigated muscles, the estimated distal-to-proximal fat fraction decline was −3.4% (0.4) (estimate [SE]) per segment ($p < 0.001$). This effect was most prominent for muscles with a mild or moderate fat fraction at baseline (i.e., 10–30% and 30–60%), for which the distal-to-proximal fat fractions declined by −4.0% (0.3) per segment ($p < 0.001$) and −9.0 (0.6) per segment ($p < 0.001$), respectively. This distal-to-proximal decrease in fat fraction across each segment was lower for muscles with <10% fat (−0.7% [0.1] per segment) and for muscles with >60% fat (−2.2% [0.4] per segment) ($p < 0.001$).

**Longitudinal changes in muscle fat fraction**. At the patient level, the longitudinal average increase in a fat fraction over all lower extremity muscles of all FSHD patients during the examination period was 4.7 ± 5.9% ($p = 0.043$), corresponding to a yearly increase of 1.3 ± 1.6%. The variability between patients was large and ranged from −0.2 to 4.5% per year. The largest progression rate was observed in the upper leg muscles (Table 1).

For individual muscles, the change in overall fat fraction depended highly on their baseline value. Fat infiltration progressed faster in muscles with a baseline fat fraction between 20–30% ($p < 0.001$), 30–40% ($p < 0.001$), 40–50% ($p = 0.016$), and 50–60% ($p = 0.048$), as compared with muscles with a baseline fat fraction of <10% (Fig. 3). The largest fat fraction increase per year occurred for muscles with a baseline fat fraction of 30–40% with a rate of 4.9 ± 2.7% per year.

We then assessed the time-dependent changes in the fat-infiltrating front (slope of the reversed sigmoid curve) along the proximo-distal axis. This assessment revealed that the fat-infiltrating front had progressed towards the proximal end of the muscles during the interscan period (see Fig. 4). This observation was further confirmed by a quantitative analysis of the change in fat fraction per segment in all analysed muscles (Fig. 5). This revealed that for muscles with low baseline fat fractions (0–20%), fat fraction increases in the most distal part of the muscle, but not in the proximal segments. The change in fat fraction then moves in a wave-like manner to the proximal end with the highest increase in fat fraction at baseline fat fraction of 20–40%, after which the change in fat fraction infiltration slows down until muscles have a fat fraction of more than 70% (Fig. 5 and Supplemental Fig. S1).

**Exceptions to the prevailing distal-to-proximal fat patterns**. Of the 386 studied muscles, 158 muscles (40.4%) could be classified as having a decreasing fat fraction in the distal-to-proximal direction. This fat infiltration pattern was best observed in the 170 mild-to-moderately fat-infiltrated muscles (fat fraction between 10–60%), as 114 of these muscles (67.1%) showed this clear decreasing fat fraction in the distal-to-proximal direction. Of the remaining 56 mild-to-moderately fat-infiltrated muscles, 28 muscles (16.5%) with an overall fat fraction below 18% showed similar fat fractions over the whole muscle length, suggesting that these muscles have a normal fat content (Supplemental Table 2). This leaves 28 muscles (16.5% of all affected muscles) that were an obvious exception to the commonly observed FF profile (Supplemental Table 3). Four of the affected muscles (one peroneus, one extensor digitorum, one vastus medialis and one rectus femoris) showed a true counter-gradient, with higher fat

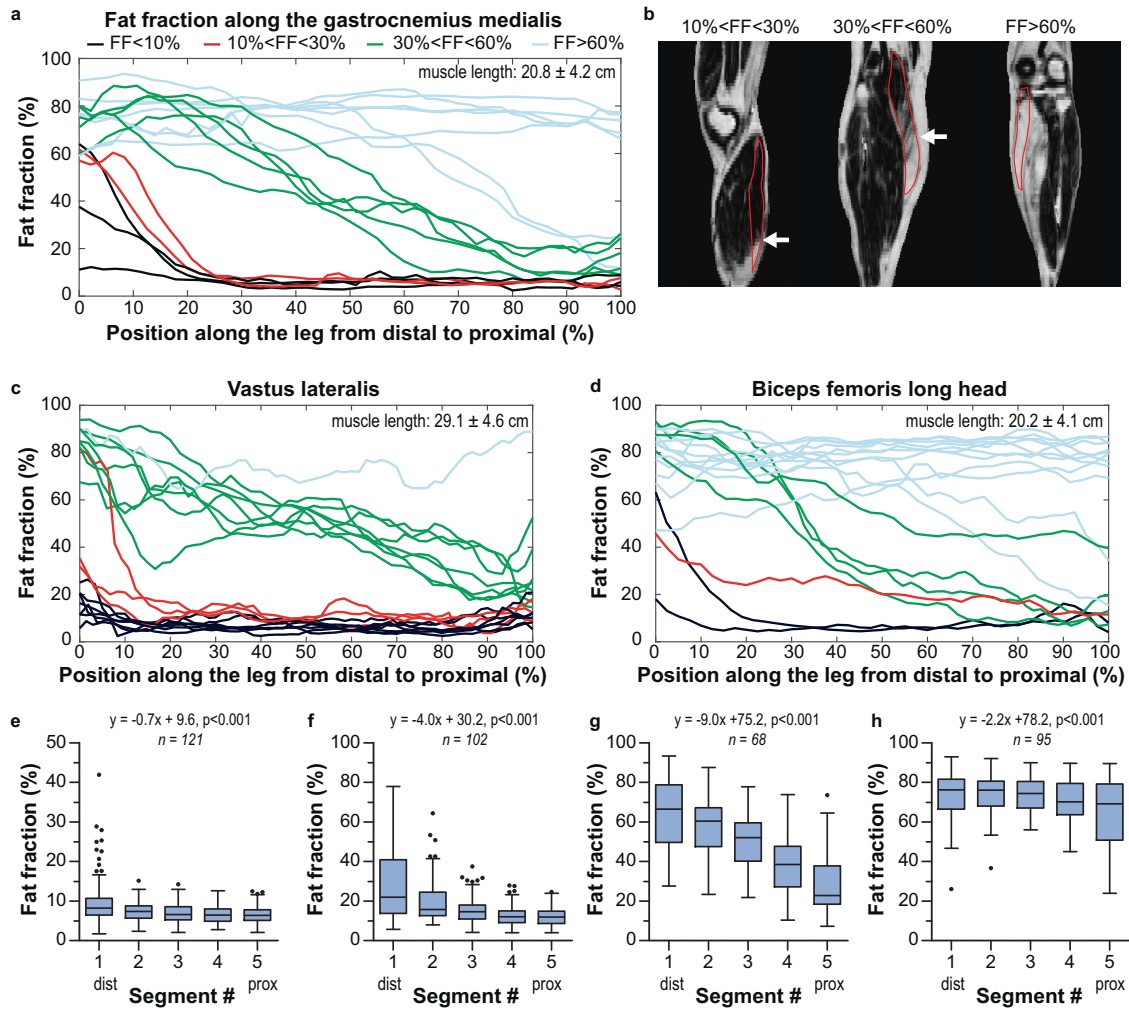

**Fig. 2 Quantitative evaluation of fat fractions along the muscle length in facioscapulohumeral muscular dystrophy (FSHD) patients. a** Fat fraction displayed at each position along the gastrocnemius medialis expressed as a percentage of muscle length from the most distal segmented slice (0%) to the most proximal segmented slice (100%). Muscles from all participants were divided into four groups: fat fraction <10% (black), 10–30% (red), 30–60% (green), and >60% (light blue). **b** Dixon fat fraction maps showing typical examples of fat infiltration in the gastrocnemius medialis (delineated in red). Left: sagittal view of the right lower leg of a 52-year-old male showing fat infiltration at the distal muscle end (white arrow depicts fat front). Middle: coronal view of the right lower leg of a 67-year-old female. The distal half of the gastrocnemius is fully infiltrated with fat, while the proximal half shows almost no fat infiltration (the white arrow depicts the fat front). Right: coronal view of the right lower leg of a 69-year-old male exhibiting a fully fat-infiltrated gastrocnemius medialis. **c**, **d** As figure **a**, but now for two upper leg muscles, vastus lateralis (**c**) and biceps femoris long head (**d**). **e**–**h** Tukey boxplots (showing median, interquartile range, and outliers) of fat fractions in five segments (distal-to-proximal) along the muscle with the corresponding linear mixed model fit for muscles with baseline fat fraction <10% (**d**), 10–30% (**e**), 30–60% (**f**), and >60% (**g**); *n* is the number of included muscles.

fractions proximally (Fig. 6a). There were also some highly fat-infiltrated semitendinosus muscles (>50% fat) showing this reversed pattern; in these muscles, it was unclear whether this was a genuine result or whether it resulted from difficulties in the manual segmentation step of these small, highly fat-infiltrated muscles. In addition, the anterior muscle compartment often showed a higher fat fraction in the most proximal slices. Occasionally, an apparent U-shaped fat pattern was observed (Fig. 6b), and several intermediately fat-infiltrated muscles displayed a homogeneous fat fraction over the whole muscle (Fig. 6c). In addition, of the affected muscles, one semimembranosus, one adductor longus, one gracilis and one vastus medialis showed a medial fat bulk (Fig. 6d).

**The effect of localised sampling on the estimated change in fat fraction.** Fat infiltration in leg muscles is commonly assessed by MRI using only centrally placed transversal slices. To evaluate

how the proximo-distal fat gradient in muscles of FSHD patients can affect these assessments for the baseline and longitudinal changes in fat fractions, we compared fat fractions calculated from MR images spanning the whole muscle with those calculated from only the central five slices (covering 2.5 cm) of each muscle and from five slices equally spread along the muscle length, i.e., a slice placed at 10%, 30%, 50%, 70%, and 90% of the muscle length. The error, i.e., the fat fraction of five slices minus the fat fraction of the whole muscle, was calculated per muscle. The significance of the absolute error, i.e., the absolute value of this difference, was tested with a linear mixed model and displayed as the estimated absolute error (SE).

For the five middle slices, the Bland–Altman analysis showed an average error of 1.4% for the baseline fat fraction and limits of agreement from −9.2 to 12% (Fig. 7a). The estimated absolute error in the baseline fat fraction was significant, being 3.7% (0.5; $p = 0.001$). The error was largest in muscles with an intermediate

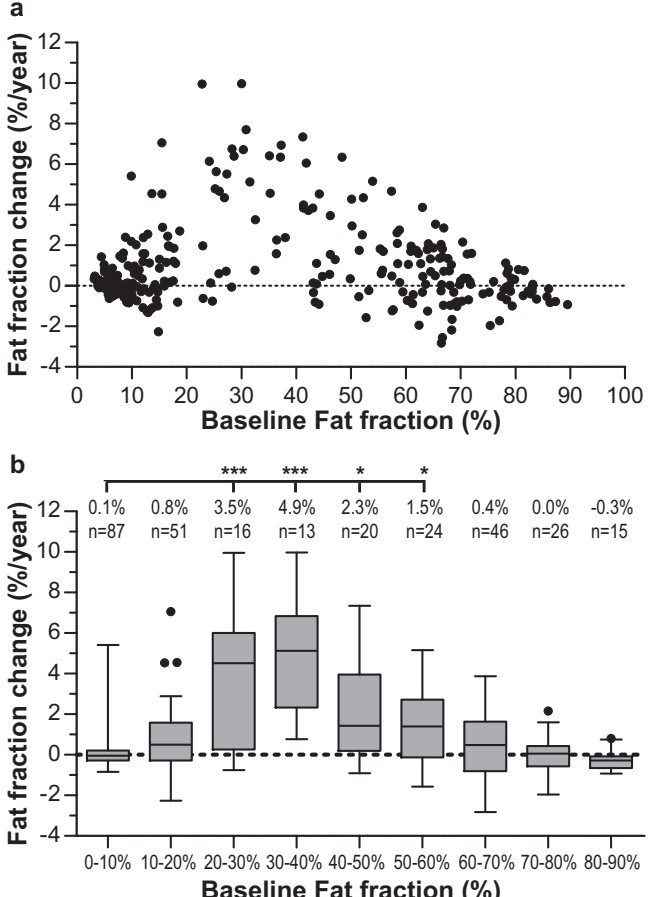

**Fig. 3 Change in fat fraction per year of muscles in facioscapulohumeral muscular dystrophy (FSHD) patients. a** Scatterplot of baseline fat fraction vs. change in fat fraction per year. Each dot represents one muscle. **b** Tukey boxplots (showing median, interquartile range, and outliers) per 10% baseline fat fraction group, indicating a faster progression in muscles with a baseline fat fraction between 20–60% compared with muscles with a baseline fat fraction of <10%. The mean values of fat fraction change and the number of analysed muscles (*n*) are displayed in the text above the boxes. *p < 0.05, ***p < 0.001.

fat fraction, where the error varied from −11.8 to +23.5%. However, when we selected five slices equally spread over the whole muscle length, the average error was reduced to 0.1% with limits of agreement from −1.5 to 1.7% (Fig. 7a). The estimated absolute error in the baseline fat fraction was reduced sevenfold, to only 0.6% (0.1; *p* < 0.001). With this approach, the error also becomes independent of the baseline fat fraction.

The change in fat fraction was also biased when only five central slices were assessed. Bland–Altman analysis revealed an average error of −0.1% per year and limits of agreement from −1.6 to 1.4% per year (Fig. 7b). The corresponding estimated absolute error was 0.6% per year (0.08; *p* = 0.001). This effect was again most prominent in intermediately fat-infiltrated muscles, where the maximum absolute error was 3.2% per year, with a large variation between individual muscles (Fig. 7c). On average, the change in fat fraction was overestimated for muscles with a baseline fat fraction ranging from 20 to 40% and underestimated for muscles with a larger baseline fat fraction (i.e., ranging from 40 to 60%) (Fig. 7c, d). These errors in the change in fat fraction estimation were reduced when five slices evenly distributed over the whole muscle length were analysed (Fig. 7b–d). This leads to an average error of 0.0% per year and limits of agreement from

−0.5% per year to 0.5% per year, with errors being independent of baseline fat fraction. The estimated absolute error was 0.2% per year (0.03; *p* = 0.001).

**TIRM hyperintensity**. Of the 396 muscles studied at baseline by TIRM imaging, 215 muscles (54.3%) were scored as TIRM positive. To assess at what muscle position these TIRM lesions occurred, the location of the lesions was visually assessed for all TIRM-positive muscles with a baseline fat fraction <30% fat (*n* = 92 muscles). This revealed that distinct TIRM lesions were most commonly present at the distal part of these muscle (59.8%) or hyperintensity lesions occurred more diffuse from the distal end extending into the whole muscle (23.7%) (Fig. 8). Only 11.3% of the distinct TIRM-positive lesions occurred at the middle part of the muscles, and 3.1% at the proximal muscle end, while the remaining 2.1% occurred at both the distal and proximal muscle end with apparent normal signal intensities seen at the muscle belly.

Furthermore, it was observed that TIRM-positive muscles are more likely to present with a distal-proximal declining fat content compared to TIRM-negative muscles (59.9% of 212 muscles vs. 23.0% of 174 muscles). Most muscles with a lower than 10% fat fraction showed no TIRM lesions (55.2%) and they contain only 16.0% of the TIRM-positive lesions.

For the follow-up MR examinations, 308 muscles were examined, of which 181 muscles (58.7%) were TIRM positive at baseline and 194 muscles (63.0%) were TIRM positive at follow-up. Most of these muscles were TIRM positive at both time points (*n* = 163, 90.1%).

## Discussion

In this MRI investigation of lower extremity muscles in FSHD patients, we quantitively assessed muscular fat infiltration in whole muscles from the distal to the proximal end, instead of only the central part of the muscle, as is common in quantitative MRI studies of diseased skeletal muscles. In this way, we demonstrate that, with a few exceptions, fat levels were highest at the distal end and lowest at the proximal end of the affected muscles. In particular, in muscles with an overall fat content between about 10 and 60%, the pattern of fat infiltration occurred as a (shifted) reversed sigmoid curve for which the slope (fat-infiltrating front) is more shallow at the higher fat content. Our longitudinal study revealed that these fat-infiltrating fronts moved to the proximal end of the muscle over the 3.5-year follow-up period. The fastest progression in the overall fat fraction was observed for muscles with mild-to-moderate fat infiltration, reaching a maximum rate of 4.9% per year in muscles with a 30–40% baseline fat fraction. Considering that muscular fat replacement is the ultimate result of cell death due to the expression of DUX4, we conclude that the disease process in FSHD starts at the distal end of leg muscles and proceeds at variable speed towards the proximal end. This conclusion is corroborated by our finding that TIRM-positive lesions, which are associated with disease activity[10,17,21,24], preferentially occur at the distal end of affected muscles in the early stage of disease development in lower extremities of FSHD patients.

An important consequence of the results of our whole-muscle analysis is that the common practice in quantitative muscle MRI of assessing only a few slices at the centre of muscles may lead to large under- or overestimations of the fat fraction and its progression. In FSHD, this under- or overestimation varied from −11.8 to +23.8% for the baseline fraction and reached up to 3.4% for the fat fraction progression rate. This error can be minimised by analysing multiple transversal slices (5 or more) evenly distributed over the whole length of the muscles. In our study, semi-automatic muscle segmentation was essential for an efficient analysis of the whole muscles[20].

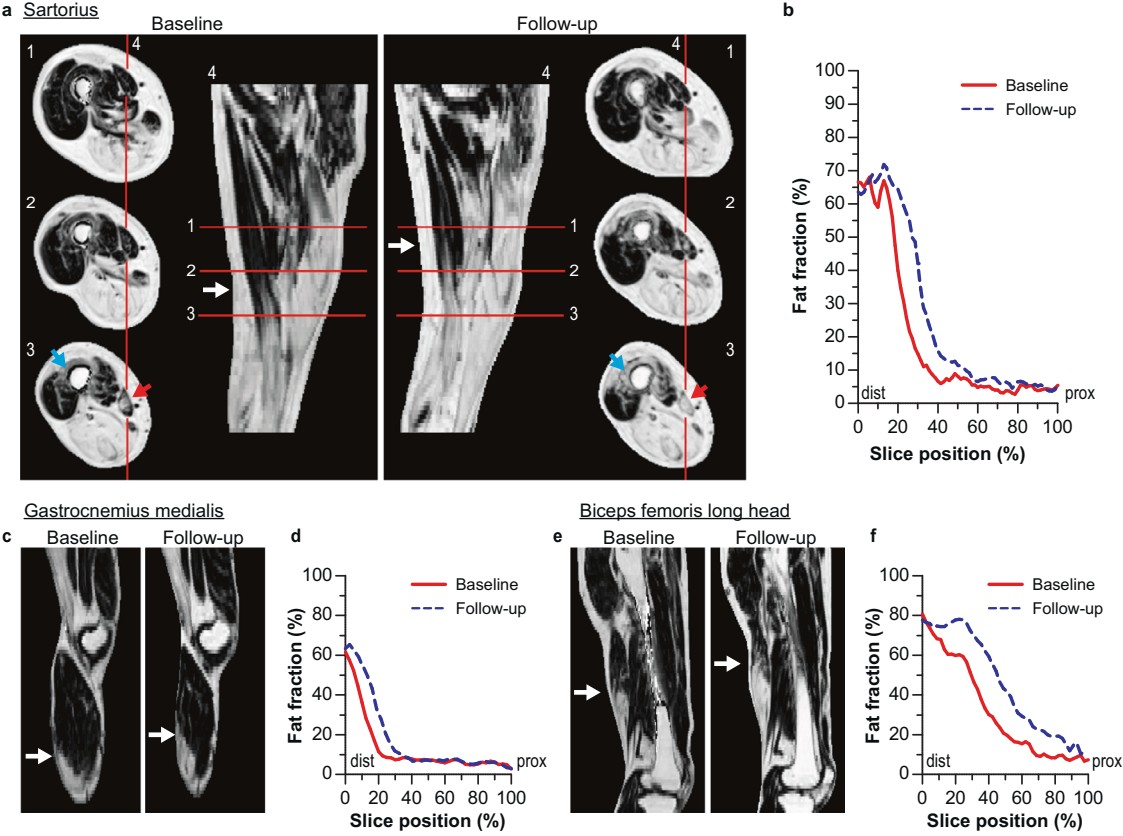

**Fig. 4 Typical examples of the proximo-distal progression of the fat-infiltrating front in muscles of facioscapulohumeral muscular dystrophy (FSHD) patients. a** Axial and sagittal cross-sections of the right upper leg of a 47-year-old male FSHD patient showing fat fraction maps at baseline and obtained after 3.8 years follow-up. They illustrate the progression of fat infiltration from distal-to-proximal in the sartorius (red arrows) and vastus intermedius (blue arrow). The position of the fat-infiltrating front for the sartorius is depicted with a white arrow in the sagittal view for baseline and follow-up. **b** Graphical display of quantitative fat fraction curve along the sartorius displayed in figure a for baseline (solid, red) and 3.8 years later (dashed, blue). **c** Sagittal view of fat fraction maps of the leg of a 52-year-old male FSHD patient at baseline and after ~3.8 years. The white arrow indicates the fat front in the right gastrocnemius medialis. **d** Graphical display of quantitative fat fraction curve along the gastrocnemius medialis displayed in figure **c** at baseline (solid, red line) and 3.8 years later (dashed, blue). **e** Sagittal view of fat fraction maps of the upper leg of a 66-year-old male FSHD patient at baseline and 3.7 years later. The white arrow indicates the fat front in the right biceps femoris long head. **f** Graphical display of quantitative fat fraction curve along the biceps femoris long head displayed in figure e at baseline (solid, red) and 3.7 years later (dashed, blue).

To the best of our knowledge, this study is the first to quantitatively follow muscular fat fraction in FSHD patients along the whole muscle length, from the distal to the proximal end. The results of the cross-sectional part of this study, showing that affected muscles in the lower extremities of FSHD patients had higher fat fractions in the distal than proximal direction, are in line with earlier studies using quantitative T2 MRI for fat fraction assessment[10,16] and reports using DIXON MRI for this assessment[13,17]. However, all these studies did not assess whole muscles end-to-end, so it remained unclear whether the proximal end of the muscle also had a relatively high-fat content and whether high-fat levels extended to the far distal end in the diseased state. Our data conclusively demonstrates that the fat fraction at the proximal muscle end is not higher than in more distal muscle parts. The findings strongly suggest that fat infiltration appears with a profile resembling a (shifted) reversed sigmoid curve and proceeds in a wave-like manner with variable speed from distal-to-proximal, with a slope (fat front) becoming more shallow during disease progression. These findings contrast previous suggestions that fat infiltration occurs via a quasi-linear gradient along the length of the muscle, a conclusion which can now be ascribed to the limited muscle coverage evaluated in these studies[10,16]. The occurrence of high-fat fractions starting distally and extending proximally during disease progression is typical for FSHD and differs from Duchenne muscular dystrophy, for which the highest fat fractions in the lower extremity muscles are found near both the distal and proximal ends, and the lowest at the muscle belly[25].

It is remarkable that the majority of affected lower extremity muscles evaluated in the present study showed the typical pattern of fat infiltration, as described above. This implies that the far distal end of leg muscles in FSHD patients is a prime location for disease initiation, which is further supported by the preferential distal occurrence of TIRM lesions early in the disease development of a muscle. One might wonder why the distal end of these muscles is more susceptible to disease initiation than more proximal parts despite the wide variation in anatomy and function between muscles. The progression of fat infiltration also raises questions about processes underlying the spread of the disease towards the proximal end of the muscle.

Recent studies indicated that, after initial bursts of DUX4 expression in a sentinel myonucleus, the DUX4 proteins diffuse into neighbouring myonuclei and participate in the activation of *DUX4* target genes[7,18,19]. In turn, the proteins of these genes activate further transcriptional factors leading to a transcriptional cascade, which is ultimately cytotoxic for myotubes. This gradually progresses along the whole muscle, consistent with a moving fat front. However, what initiates DUX4 bursts is still

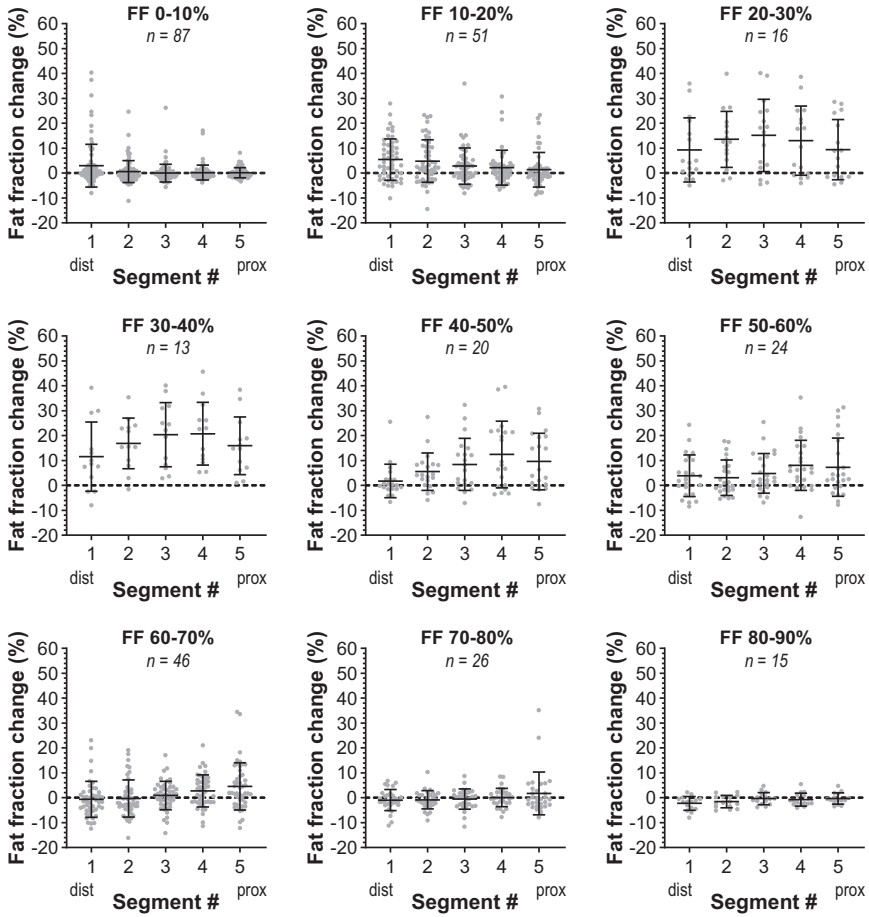

**Fig. 5 Wave-like progression of fat infiltration along the proximo-distal muscle axis in facioscapulohumeral muscular dystrophy (FSHD) patients.** Each panel represents muscles with a 10% incremental range of baseline fat fraction (FF) values. At a baseline fat fraction <10%, only a marginal change in the average fat fraction is seen for the first most distal segment. At a baseline fat fraction of 10–20%, the average fat fraction increases in the distal segments (segments 1 and 2). At a baseline fat fraction of 20–40%, the average fat fraction increases in all segments, but most in the middle three segments (segments 2, 3, and 4). At a baseline fat fraction of 40–80%, the average fat fraction increases, especially in the proximal segments (segments 4 and 5), but the increase is lower than that for fat fraction between 20–40% and steadily diminishes at a higher baseline fat fraction. *n* is the number of included muscles. Data were presented as mean ± SD and individual data points were displayed as grey dots. Fat fraction changes normalised to residual muscle mass are displayed in the supplemental note.

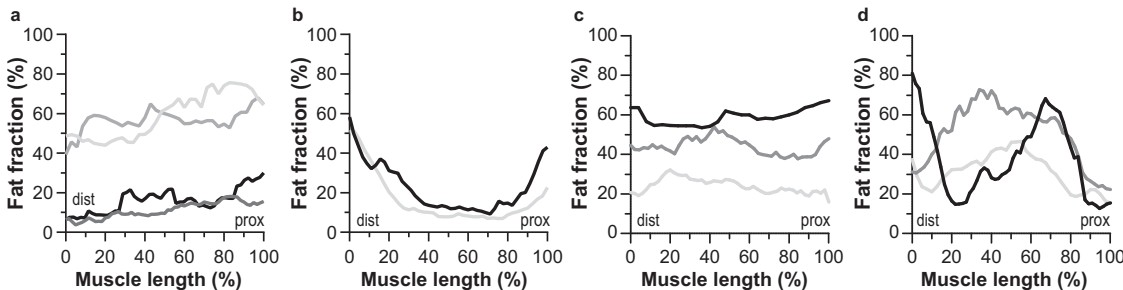

**Fig. 6 Typical examples of muscles in facioscapulohumeral muscular dystrophy (FSHD) patients not showing a decreasing distal-to-proximal fat infiltration profile. a** One vastus medialis (light grey), one rectus femoris (medium grey), one peroneus (dark grey), and one extensor digitorum (black) muscles showing a reversed gradient, with a higher fat fraction proximally. **b** Tibialis anterior muscles showing a higher fat fraction distally and proximally. **c** Muscles with a homogenously elevated fat fraction along the muscle: peroneus (light grey), extensor digitorum (dark grey), and biceps femoris short head (black). **d** Muscles with fat infiltration in the middle of the muscle: gracilis (light grey), vastus medialis (dark grey), and rectus femoris (black).

unknown, although studies on myoblasts from FSHD patients suggest that these bursts occur stochastically[26,27]. Our results indicate that specific conditions at the distal end of leg muscles are involved and, thus, that initiation is not purely stochastic. In Duchenne muscular dystrophy, it has been suggested that mechanical strain on the tendons contributes to disease

initiation[25], but in boys with this dystrophy, fat infiltrates muscles at both ends, in contrast to what we observed in muscles of FSHD patients.

The physiology of muscles may vary substantially along their length. Recently, the tibialis anterior was shown to have a relatively low oxidative capacity and post-exercise $O_2$ supply in the

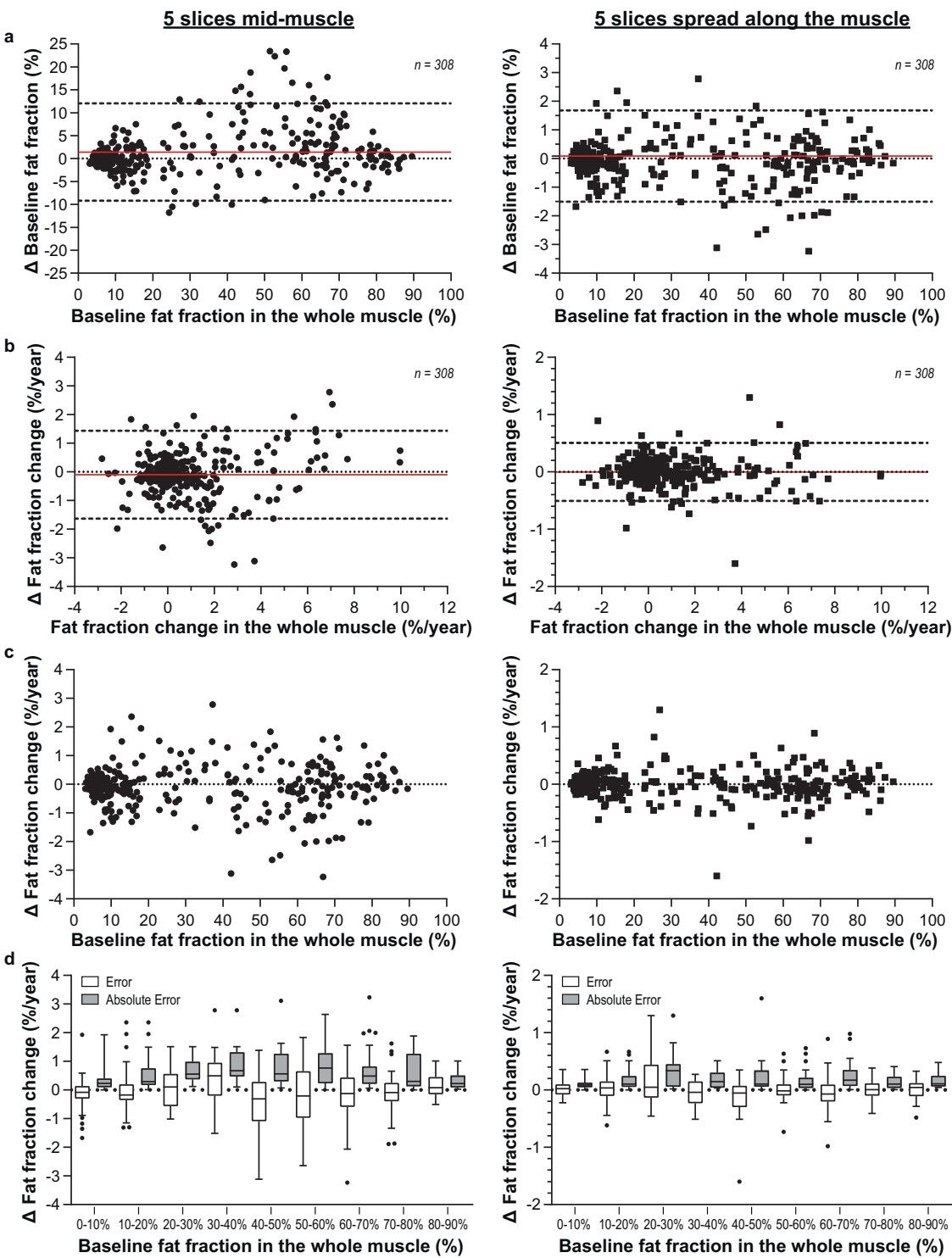

**Fig. 7 Effect of localised sampling on the estimated fat fraction and its progression in muscles of facioscapulohumeral muscular dystrophy (FSHD) patients. a** Bland–Altman graph displaying the error (Δ) in baseline fat fraction due to localised sampling of five slices in the middle (left) and spread along the muscle length (right) against the baseline fat fraction of the whole muscle (error = fat fraction of five slices - fat fraction of all slices). Each dot/square represents one muscle. Average error and limits of agreement are indicated by the red line and dashed lines, respectively. **b** Bland–Altman graph displaying the error in a change in fat fraction per year due to localised sampling of five slices in the middle (left) and spread along the muscle length (right) against the change in fat fraction in the whole muscle (error = change fat fraction of five slices - change fat fraction of all slices). **c** Error in change in fat fraction displayed against the baseline fat fraction of the whole muscle. **d** Tukey boxplots (showing median, interquartile range, and outliers) of the error (white) and absolute error (grey) for individual muscles, with the muscles grouped based on whole-muscle baseline fat fraction (10% increments). When analysing only the five central slices (left), the error (grey bars) shows that the change in fat fraction is generally overestimated in muscles with a baseline fat fraction of 20–40% and underestimated in muscles with a baseline fat fraction of 40–60%. The white bars show that in intermediately fat-infiltrated muscles (10–60% baseline fat fraction), the absolute error was about 0.8%/year. This absolute error is significantly reduced when the five slices are spread along the muscle length (right), to only 0.2%/year. *n* is the number of analysed muscles.

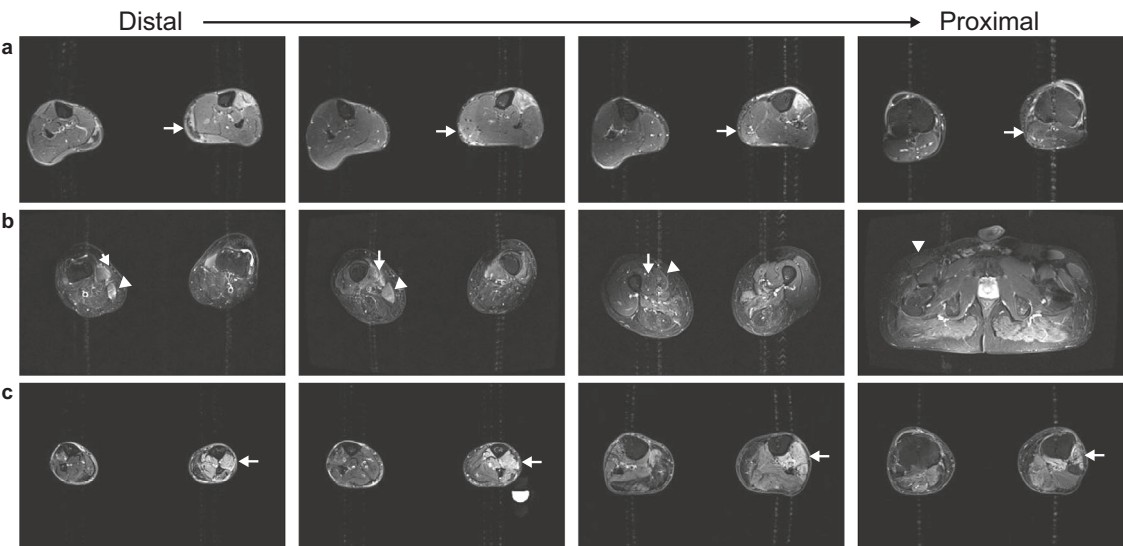

**Fig. 8 Typical examples of hyperintensity patterns along the muscle length at a turbo-inversion recovery magnitude (TIRM) sequence.** The order of the images from distal-to-proximal is indicated at the top of the figure. **a** Lower legs of a 52-year-old male facioscapulohumeral muscular dystrophy (FSHD) patient showing TIRM hyperintensity in the distal part of the gastrocnemius lateralis and a normal appearance of this muscle at the proximal muscle end (arrows). **b** Upper legs of a 47-year-old male FSHD patient with TIRM hyperintensity localised in the distal part of the sartorius (arrow heads) and vastus medialis (arrows) with a normal appearance at the proximal end of these muscles. **c** Lower legs of a 62-year-old male FSHD patient showing TIRM hyperintensity over the whole muscle length for all lower leg anterior compartment muscles (arrows).

distal part of the muscle[28,29]. This distally lower post-exercise $O_2$ supply also occurs in the quadriceps and gastrocnemius[30–33]. Interestingly, oxidative stress in FSHD-affected muscle cells is implicated in disease onset by increasing DUX4 expression[34]. Moreover, DUX4 activates oxidative stress responses, leading to cell death[35–37], and affected muscle cells are vulnerable to oxidative stress[38,39]. Therefore, it is of interest to know if (transient) hypoxia at the distal end of FSHD-affected leg muscles plays a role in disease initiation. Whatever the molecular conditions facilitating disease onset, our results provide a framework for further investigation into the nature of this process.

The overall change in the fat fraction of 1% per year indicates that fat progression in the lower extremities of FSHD patients at the individual level is slow. This percentage change is relatively low compared to those observed in other FSHD studies, which ranged from 0.86% per year to 5.4% per year[10,14,15,24,40]. This large variability among reported progression rates in a given patient cohort is not surprising, given that the rates are affected by multiple factors such as muscle type, the investigated location along the proximo-distal axis, the disease severity at baseline, and the applied MRI technique. For example, our study and that of others showed that fat infiltration in upper leg muscles progresses faster as compared to lower leg muscles[14,15]. Moreover, previous studies have assessed mainly central muscle areas, which can lead to over- or underestimation of fat progression (vide infra), which also depends on baseline fat fraction. Furthermore, more rapid fat infiltration rates have been reported in the muscles of FSHD patients involved in active disease processes, i.e., those showing TIRM-positive foci or those in a state of intermediate fat infiltration[24]. Finally, methodological differences, such as different MRI acquisition methods (e.g., 2 pt Dixon, multi-echo spin-echo) and data processing tools, may have slightly different sensitivities in the quantification of fat fractions. For example, the 2pt-Dixon study used in our study is likely to be less accurate than a multi-point Dixon, however, it does not affect precision and this precision is what really matters in longitudinal studies.

Despite that the leg muscle fat infiltration rate at the participant level was slow, fat infiltration can progress quickly in

individual muscles. The fat replacement rate rapidly accelerates up to an average of 4.9% fat per year following fat infiltration onset until a muscle's fat fraction becomes about 40%. At this stage, the fat replacement rate decelerates until muscles are completely fat-infiltrated. This typical pattern is in line with other longitudinal studies conducted in FSHD patients[10,14,15]. The pattern also fits the clinical picture reported in FSHD patients, with long stable periods alternating with short periods of rapid deterioration in single muscles or muscle groups[41]. In other muscular dystrophies, the relationship between the baseline fat fraction and the change in fat fraction has been studied in less detail. In late-onset Pompe disease, myotonic dystrophy type 1, and spinal muscular dystrophy, the change in fat fraction in muscles with elevated baseline fat fraction was larger than in muscles with fat fractions in the range of healthy controls[42–44].

The presence of a distal-to-proximal fat-infiltrating front in diseased muscles has major implications for the application of quantitative MRI as a biomarker to assess treatments in clinical trials[6,45,46]. We demonstrated that evaluating fat fractions only in the central muscle parts, a commonly applied approach, instead of considering the whole muscle length, led to an average absolute error of 3.7% in baseline fat fraction and 0.6% per year in fat replacement rate in lower extremity muscles. The latter estimated absolute error is very large, representing 50% of the total change in fat fraction over 1 year commonly reported[15,24]. For instance, even if clinical trials intend to monitor disease progression in intermediately fat-infiltrated leg muscles—which are known to be most sensitive to changes—the corresponding evaluation of baseline fat fraction and fat replacement rate may both be largely biased if measurements are performed for slices located away from the fat progression front. Such an observation was made in a study of muscles in Duchenne muscular dystrophy, where a slight shift of 1.5 cm along the proximo-distal muscle axis led to a significant difference in the reported fat fraction[25], especially in intermediately fat-infiltrated muscles. The absolute errors in baseline fat fraction and change in fat fraction can be substantially reduced when the five analysed slices are distributed along the muscle length. These findings demonstrate that whole-muscle

imaging is essential to properly diagnose disease severity and to assess disease progression and response to therapy, such as in clinical trials. However, given that manual segmentation in quantitative assessments of whole muscle is very time-consuming, automatic or semi-automatic muscle segmentation approaches are critical in clinical assessments[47]. The analysis of an adequate number of slices properly divided over the length of muscles may represent a valid alternative.

In conclusion, fatty replacement in affected lower extremity muscles of FSHD patients commonly starts at the distal end of the muscles. It progresses relatively rapidly, in a wave-like manner, in the proximal direction in the mid-phase of disease development, after which it proceeds at a slower pace towards the proximal muscle end. In the early phase of disease development, TIRM-positive lesions dominantly occur at the distal end of affected muscles. These observations identify the distal end of lower extremity muscles as a prime location for disease initiation and are consistent with proposed mechanisms for disease progression in muscles. Factors involved in the activation of DUX4 expression to initiate disease have yet to be discovered, but oxidative stress at the distal end of leg muscles may play a role. Our analysis of whole muscles in FSHD demonstrates that the common practice in muscle MRI studies of assessing only a few slices in more central parts of muscles may lead to large under- or over-estimations of muscle fat fractions and their change over time.

## Data availability

Source data for the figures are available as Supplementary Data 1. All post-processed data generated or analysed during this study are included in Supplementary Data 2. All other data were available from the corresponding author upon reasonable request.

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

## Acknowledgements
This work was funded by Friends of FSH Research (awarded in 2018) and the initial phase of this project was supported by aTyr Pharma. We would like to thank our participants and the research assistants for their help in contacting the participants. Furthermore, we thank Sanne Vincenten for her help with comparing data acquired before and after the MR system upgrade and Marieke Ploegmakers for scoring the TIRM images.

## Author contributions
The experiments were performed by L.H. at the department of Medical Imaging/Radiology of the Radboud university medical center, Nijmegen, The Netherlands. L.H. and A.H. were involved in the conception or design of the work. A.O. developed the semi-automatic segmentation software under the supervision of D.B. and applied it to the FSHD data of this study. L.H. did further data processing and wrote the first draft. All authors were involved in revising the manuscript.

## Competing interests
The authors declare no competing interests.
