## [Peer Review File · Communications Medicine]

Reviewers' comments:

Reviewer #1 (Remarks to the Author):

The present paper reports quantitative MRI data on lower-limb muscles from FSHD patients, with the aim to evaluate the progression profile of the disease along the full proximal-distal axis of muscles. The main finding of the work confirms previous literature observations of a disto-proximal gradient of involvement in this disease; the added value here is the rigorous quantitative MRI pipeline used (acquisition, segmentation and analysis) that partially overcome the limited number of patients studied. The results on the comparison between whole muscle vs central slices assessment are relevant, too. They point out a potentially important source of error in the use of quantitative MRI as outcome measure in clinical trials on this disease.

I have a couple of minor comments:

- Do Authors have data on STIR or wT2 of the studied muscles? If so, it would be interesting to compare them with the findings on fat infiltration distribution and progression.
- Discussion [307-309]: I invite authors to underlie the limitation of 2-pt DIXON sequences
- Discussion [273-295]: this paragraph seems speculative and redundant, I suggest to shorten it

Reviewer #2 (Remarks to the Author):

The authors have investigated a small number of FSHD patients and have measured the fat replacement along the length of muscles in the lower limbs. They confirm the distal to proximal progression of skeletal muscle destruction which was described 15 years ago (See Kan et al) and provide with a refined description of the muscle fatty replacement features in this disease.

The data were carefully analyzed, are clearly presented and the whole paper is well written.

The authors think they have identified a wave-like progression of fat replacement as opposed to a linear gradient progression which was previously suggested. They present some data showing a wave like progression of FF in figure 4. However the figure 5 which summarizes all observations is not supportive of this and displays unambiguously a gradient of FF fraction along the muscle length and a very similar progression of the fatty replacement over time. See panels 20-30%, 30-40%, 40-50%.

One of the significant contributions of the present study is to identify muscles that do not follow this distal to proximal progression.

There are questions on how the scaling of muscles was performed. How were the 0 and 100% landmarks determined ? See specific questions below.

As understood from the methods, the water and fat components were separated with a standard Dixon reconstruction, known to underestimate the fat fraction and with a variable impact using different echo times (Trio vs Prisma). A known improvement is to use a 5 or 6 lipid resonance model with fixed relative amplitudes. The authors might want to use this more advanced processing, available in several laboratories (Leiden, Paris, Newcastle). It might avoid the cosmetic correction

applied to the transition from Trio to Prisma.

The authors spent a great amount of time in demonstrating that the FF increase is the fastest in muscle with intermediary fatty replacements and slows down at higher FF. It is inherent to the way the FF progression is expressed and also found in other diseases, for example the Duchenne muscular dystrophy. What is already fatty replaced cannot be fatty replaced. A more exact way to evaluate the disease activity is to normalize the FF increase to the residual muscle mass to obtain a true muscle fatty replacement rate ($=\Delta FF/t \cdot (1 - \text{initialFF})$). This index is much less popular than the standard fatty progression rate ($=\Delta FF/t$). The authors might be tempted to use this index too, which might bring interesting observations on the disease activity along the muscle length.

There is no data, qualitative or quantitative, on the disease activity as it can be estimated with STIR T2w or multi TE SE acquisitions. There are particular important in FSHD, where several studies have shown a relation between a relatively sudden increase in T2 and the rapid subsequent muscle destruction. Why have water T2 measurements not been acquired? Or if acquired not integrated in this paper?

P4 I91 and followings : what was the slice thickness (or stack length?)

P4 I103 : The relationship between the fat fraction before and after the upgrade fit a linear model (baseline fat fraction corrected = $1.1864 \times \text{baseline fat fraction} - 2.7878$). What is the r^2 of the relation?

P5 I113-114 : Subsequently, a composite fat fraction of the whole lower extremity of each subject was estimated by taking the average of the fat fractions of the individual muscles. Was it the simple average of the individual FF or was it the weighted average by muscles CSA or volumes? The second approach is the right one. Please clarify and possibly amend.

P5 : what anatomical landmarks were chosen to determine the muscle extremities? How was the muscle length taken into account to determine and compare the FFs between individuals with different limb lengths? In other words, how was the segment location determined? As a percentage of the muscle lengths? or in cm? from which anatomical landmark?

P6 I142 and followings : « To estimate the size of this effect in FSHD, we compared the baseline fat fraction and change in fat fraction over the five middle slices to the whole muscle analysis using a Bland–Altman analysis. Furthermore, we calculated the absolute difference in baseline fat fraction and change in fat fraction over the middle five slices compared with the whole muscle ». Knowing the uneven distribution of FF along the muscle length, it would have been fair to also compare measures performed on only 5 slices but more separated on the muscle length.

P7 I173 : « The presence of a fat gradient with the highest fat fraction distally was further demonstrated by quantitative assessment of the five muscle segments (Fig 2C–F). » I did not read how the segments were defined. Please clarify.

P8 I218 : what was the length covered by the 5 adjacent slices?

Figure 2 : how was the position along the leg calculated in %? Was the slice thickness (stack length) adjusted to the leg length. If the slice thickness was fixed, how were the FF transformed in FF in % along the leg muscle. Also, as already asked, what were the anatomical landmarks chosen for 0 and 100%.

Figure 4: nice examples of the front progression of fatty replacement but in Figure 5 : no evidence of such front progression at fatty replacement.

Reviewer #3 (Remarks to the Author):

The article “Whole-muscle fat analysis identifies distal muscle end as disease initiation site in facioscapulohumeral muscular dystrophy” is a solid contribution to the muscle MRI literature on FSHD. Its main focus is on seven male patients and 298 of their leg muscles longitudinally over 3.5 years. The main novel findings are that 1) distal muscle are the most affected and 2) how we sample muscle slices affects our summation of fat fraction as the fat is distributed unequally across the muscle.

This brings into attention that our prior ways of averaging only the central slices subjects our analyses to more variability; and is helpful to future analyses that we should calculate more slices.

The methods for acquisition and analysis of the muscle MRI were sound. The results and graphs are well presented, specifically Figures 2-5 are especially helpful to push forward the field. The discussion was reasonable and well written. Some minor points:

1. Describe more the muscles that were exception to the rule of the distal-to-proximal gradient. 43% had clear decreasing fat fraction; 34% of muscle (low fat infiltration) and 11% of all muscles (high fat infiltration) had no or minimally decreasing fat fraction. Leaving 12%/36 muscles that did not follow the rule. An appendix table of those muscles, in terms of their position and amount of fatty infiltration at baseline, or more examples of those muscles not showing the gradient would be helpful (a full figure 6 of the other muscles).
2. Did the authors see small areas of T1 fat in the muscles that coalesced as the fat progressed from distal to proximal area. I.e., while there is a distal -> proximal march of fat, did the authors see small pockets of T1+ fatty infiltrated areas that would be made confluent over time.
3. The authors measured the fat fraction on every fifth slice and then coalesced the fat fraction into 5 equally distributed segment along the muscle axis. Did the authors go back and check whether the coalesced 5 segments in the baseline and follow-up MRIs corresponded to roughly the same areas, or was there also noise and inexactness in determining the 5 segments? If that occurred, does that change the findings?
4. The description of Figure 7D was slightly confusing to the reviewer and would be benefited by describing it better/more in the legend and the results section.
5. Expound more in the discussion on the limitation that this study also only samples every 5th slice (due to the time intensity of segmenting and analyzing each slice) and that ideally, sampling as many slices from the distal to proximal end would probably provide the most accurate description of the muscle fat burden. This does not affect the paper’s central posit that a distal-to-proximal gradient is present in most muscles, could affect the summed average fat fraction.

RESPONSE TO REVIEWER'S COMMENTS

We thank the reviewers for their valuable comments on our manuscript with the title: **"Whole-muscle fat analysis identifies distal muscle end as disease initiation site in facioscapulohumeral muscular dystrophy"** by L. Heskamp, A. Ogier, D. Bendahan, A. Heerschap. (COMMSMED-22-0029-T)

Our response to each comment is indicated in blue. Changes in the manuscript in response to the comments of the reviewers are labeled via comments and highlighted via tracked changes.

Response to Reviewer #1

The present paper reports quantitative MRI data on lower-limb muscles from FSHD patients, with the aim to evaluate the progression profile of the disease along the full proximal-distal axis of muscles. The main finding of the work confirms previous literature observations of a disto-proximal gradient of involvement in this disease; the added value here is the rigorous quantitative MRI pipeline used (acquisition, segmentation and analysis) that partially overcome the limited number of patients studied. The results on the comparison between whole muscle vs central slices assessment are relevant, too. They point out a potentially important source of error in the use of quantitative MRI as outcome measure in clinical trials on this disease.

Response: Thank you for these comments. We would like to point out that there is no previous literature describing a proximal-distal fat gradient in lower extremity muscles all the way from one muscle end to the other. In this sense our observations are new and in our opinion also important as the results allow us to conclude that disease is initiated at the far distal end of these muscles; a conclusion that could not have been drawn from data in previous literature.

I have a couple of minor comments:

1. Do Authors have data on STIR or wT2 of the studied muscles? If so, it would be interesting to compare them with the findings on fat infiltration distribution and progression.

Response: Thank you for pointing this out. We did indeed acquire STIR (or TIRM) images from these patients as well. We have now analysed these scans and added text about the TIRM examinations and results to the manuscript. Interestingly, the results show that in the early development of the disease in muscles TIRM positivity preferential occurs at the distal muscle end, thus supporting our conclusion that this end is the prime disease initiation site.

Added text methods

"Furthermore, edematous lesions were assessed with a Turbo Inversion Recovery Magnitude (TIRM) sequence using the following settings at both baseline and follow-up: repetition time = 4140 ms, echo time = 41 ms, inversion time = 240 ms, flip angle = 150°, field of view = 435x271 mm, in-plane resolution = 1.36 x 1.36 mm, slice thickness = 5 mm and slice gap = 5 mm. The subsequent stacks were aligned in the feet-head direction, with no overlap." (Page 4 Line 103-107)

“The TIRM images were reviewed by an experienced musculoskeletal radiologist and scored for the presence of signal hyperintensity. To determine at which position along the muscle length the TIRM hyperintensity preferentially occurred, we visually scored the location of the TIRM lesions for all TIRM positive muscles with low or mild fat infiltration (<30% fat fraction). For this purpose, each muscle was evenly divided in three parts: distal, middle and proximal.” **(Page 6 Line 142-146)**

Added text results **(Page 10-11 Line 292-309)** + added Figure 8 **(Page 32)**:

“Of the 396 muscles studied at baseline by TIRM imaging, 215 muscles (54.3%) were scored as TIRM positive. To assess at what muscle position these TIRM lesions occurred, the location of the lesions was visually assessed for all TIRM positive muscles with a baseline fat fraction < 30% fat (*n*=92 muscles). This revealed that distinct TIRM lesions were most commonly present at the distal part of these muscle (59.8%) or hyperintensity lesions occurred more diffuse from the distal end extending into the whole muscle (23.7%) (Figure 8). Only 11.3% of the distinct TIRM positive lesions occurred at the middle part of the muscles, and 3.1% at the proximal muscle end, while the remaining 2.1% occurred at both the distal and proximal muscle end with normal signal intensities seen at the muscle belly.

Furthermore, it was observed that TIRM positive muscles are more likely to present with a distal-proximal declining fat content compared to TIRM negative muscles (59.9% of 212 muscles vs. 23.0% of 174 muscles). Most muscles with a lower than 10% fat fraction showed no TIRM lesions (55.2%) and they contain only 16.0% of the TIRM positive lesions.

For the follow-up MR examinations, 308 muscles were examined of which 181 muscles (58.7%) were TIRM positive at baseline and 194 muscles (63.0%) were TIRM positive at follow-up. Most of these muscles were TIRM positive at both time-points (*n*=163, 90.1%).”

We addressed these findings in the discussion.

- “This conclusion is corroborated by our finding that TIRM positive lesions, which are associated with disease activity^{10,17,23,24}, preferentially occur at the distal end of affected muscles in the early stage of disease development in lower extremities of FSHD patients.” **Page 11 Line 325-328**
- “..., which is further supported by the preferential distal occurrence of TIRM lesions early in the disease development of a muscle.” **Page 12 Line 359-360**
- “In the early phase of disease development, TIRM lesions dominantly occur at the distal end of affected muscles.” **Page 15 Line 441-442**

These findings have also been added to the abstract **(Page 2)**

2. Discussion [307-309]: I invite authors to underlie the limitation of 2-pt DIXON sequences

One major limitation of this simple, so-called, two point Dixon approach [16] is that it assumes that the magnetic field is homogeneous everywhere, which is generally not true. However, this complication can be accommodated by acquiring a third echo

Response: The main disadvantage of the two echo approach is that it assumes a homogeneous magnetic field, which may not be the case in imaging both legs. Consequently, fat/water swaps can occur between legs and within a leg. In our study, this problem occurred only once and was dealt with by excluding the lower leg in which this fat/water swap occurred. Furthermore, the accuracy of the 2pt-Dixon is likely lower than a 3pt-Dixon. However, the precision or discriminant power is not affected, and this is what matters for longitudinal studies.

We have now recognized this limitation in the discussion

“For example, the 2pt-Dixon used in our study is likely to be less accurate than a multi-point Dixon approach, however it should not have affected the precision and this precision is what matters for longitudinal studies.” Page 13 Line 403-405

3. Discussion [273-295]: this paragraph seems speculative and redundant, I suggest to shorten it.

Response: We have written this paragraph because we believe that putting our results in the context of what is known about mechanisms involving DUX4, that underly the development of FSHD, is very relevant, not only to provide possible molecular and physiological explanations for our findings, but also as a potential framework for further research. However, we do understand that such a paragraph should not be too long and therefore we have shortened the text as suggested by the reviewer. See modified version of the manuscript. Page 12 Line 364-389

Response to Reviewer #2:

The authors have investigated a small number of FSHD patients and have measured the fat replacement along the length of muscles in the lower limbs. They confirm the distal to proximal progression of skeletal muscle destruction which was described 15 years ago (See Kan et al) and provide with a refined description of the muscle fatty replacement features in this disease.

Response: We thanks this reviewer for very valuable comments on our manuscript. We would like to emphasize that the results of our studies go far beyond a confirmation of previous work on fatty replacement concerning distal-proximal progression. Please see our comment on the general remarks of reviewer 1.

The data were carefully analyzed, are clearly presented and the whole paper is well written.

1. The authors think they have identified a wave-like progression of fat replacement as opposed to a linear gradient progression which was previously suggested. They present some data showing a wave like progression of FF in figure 4. However the figure 5 which summarizes all observations is not supportive of this and displays unambiguously a gradient of FF fraction along the muscle length and a very similar progression of the fatty replacement over time. See panels 20-30%, 30-40%, 40-50%.

Response: By taking the increase in overall fat fraction as a measure for progression of the disease in a muscle, as represented by the fat fraction groups in percentage ranges (FF<10%, 10-30%, 30-60%, >60%), the wave-like pattern of progression can also be deduced from the curves shown in Figure 2A. To further illustrate that this is a common observation among leg muscles we have added examples of end-to-end fat curves from muscles of the thigh (new Figure 2C). In fact the majority of these end-to-end fat curves resemble (shifted) reversed sigmoid curves. For Figure 5 we have averaged the data of these fat curves per fat fraction group and plotted these as a function of muscle segments. However, as the curve for each individual muscle in a fat fraction group may be (slightly) shifted (in x and y direction) with respect to the others the appearance of a wave-like progression after averaging the data is indeed not so obvious anymore. The wave-like appearance is much better visible if we plot the change in fat fraction per segment for each group as these are less influenced by fat fraction curve shifts among individual muscles. Therefore we have replaced figure 5 with a figure showing the change in fat fraction per segment. This clearly identifies a wave-like progression of fat infiltration in lower extremity muscles. Such a pattern of progression cannot be explained by fat infiltration along a linear gradient. The text in the Results and Discussion sections describing and interpreting the new figures 2 and 5 have been highlighted (Page 7/8 Line 195-216, Page 8/9 Line 229-237, Page 11 Line 317-318, Page 12 Line 336-354, and Page 15 Line 437-441).

One of the significant contributions of the present study is to identify muscles that do not follow this distal to proximal progression.

Response: Thank you very much for pointing this out.

2. There are questions on how the scaling of muscles was performed. How were the 0 and 100% landmarks determined ? See specific questions below.

Response: We like to refer the reviewer to our answers given to this reviewers question 9 and 13.

3. As understood from the methods, the water and fat components were separated with a standard Dixon reconstruction, known to underestimate the fat fraction and with a variable impact using different echo times (Trio vs Prisma). A known improvement is to use a 5 or 6 lipid resonance model with fixed relative amplitudes. The authors might want to use this more advanced processing, available in several laboratories (Leiden, Paris, Newcastle). It might avoid the cosmetic correction applied to the transition from Trio to Prisma.

Response: We appreciate this comment from the reviewer, and would have preferred to reconstruct the fat and water images as suggested. However, we only have the magnitude images of the two separate echoes available to us, and not the phase images. Consequently, we cannot perform advanced processing post-hoc.

4. The authors spent a great amount of time in demonstrating that the FF increase is the fastest in muscle with intermediary fatty replacements and slows down at higher FF. It is inherent to the way the FF progression is expressed and also found in other diseases, for example the Duchenne muscular dystrophy. What is already fatty replaced cannot be fatty replaced. A more exact way to evaluate the disease activity is to normalize the FF increase to the residual muscle mass to obtain a true muscle fatty replacement rate ($=\Delta \text{FF}/t \cdot (1 - \text{initialFF})$). This index is much less popular than the standard fatty progression rate ($=\Delta \text{FF}/t$). The authors might be tempted to use this index too, which might bring interesting observations on the disease activity along the muscle length.

Response: We thank the reviewer for these valuable comments. In our original manuscript we presented FF changes for the overall FF (i.e. total FF) per muscle (see Fig 3). Indeed in the interpretation of this representation it should be taken into account that for each overall FF at baseline the change in FF also depends on the value of this overall FF. In addition, to circumvent this issue, we determined the FF for separate muscle segments (instead of the whole muscle) in 9 overall FF groups and plotted the data at baseline and after ~3.5 years to illustrate the rate of fatty replacement (see old Fig 5). However, as outlined in our response to point 1 of this reviewer the averaging of FFs of different muscles in a group may cause blurring of the actual FF changes. Therefore we have now replaced this by an analyses of the ΔFF of muscle segments (new Fig 5), which much better demonstrates that the FF increase is indeed the fastest in muscles with intermediary fatty replacement and also that fatty replacement proceeds in a wave-like manner. In addition we have followed the suggestion of the reviewer to normalize the ΔFF to the residual muscle mass (Fig S1). The results were similar, although somewhat more pronounced, which validates our approach of analysing segments along the muscle.

5. There is no data, qualitative or quantitative, on the disease activity as it can estimated with STIR T2w or multi TE SE acquisitions. There are particular important in FSHD, where several studies have shown a relation between of a relatively sudden increase in T2 and the rapid subsequent muscle destruction. Why have water T2 measurements not been acquired ? Or if acquired not integrated in this paper?

Response: This is a relevant question indeed and is also posed by reviewer 1. Therefore, we like to refer the reviewer to our response to question 1 for reviewer 1. We have now added the analysis of the TIRM data, which shows that also TIRM hyperintensity preferentially occurs distally in the early development of FSHD in leg muscles. Multi-echo spin-echo sequences were not acquired because this was not the main focus of the research question. Adding multi-slice multi-spin echo sequence stacks that cover the whole muscle length would have made the total acquisition time too long.

6. P4 l91 and followings : what was the slice thickness (or stack length?)

Response: The slice thickness was 5 mm. This was incorporated in a 3D voxel size, but we recognize this was not very obvious from the description. Therefore, we have now explicitly added the slice thickness to the text (Page 4, Line 100)

7. P4 l103 : The relationship between the fat fraction before and after the upgrade fit a linear model (baseline fat fraction corrected = $1.1864 \times \text{baseline fat fraction} - 2.7878$). What is the r^2 of the relation?

Response: The r^2 is 0.994, we have added this to the manuscript (Page 5 Line 117)

8. P5 l113-114 : Subsequently, a composite fat fraction of the whole lower extremity of each subject was estimated by taking the average of the fat fractions of the individual muscles. Was it the simple average of the individual FF or was it the weighted average by muscles CSA or volumes ? The second approach is the right one. Please clarify and possibly amend.

Response: For the composite fat fraction we determined the simple average of the individual fat fraction or change of fat fraction per muscle. We have now recalculated our outcome measures and have taken the weighted average by muscle volume. This leads to slightly different values, but the main outcomes did not change.

The manuscript has been changed at the following positions

- Page 5 Line 127-129: Methods - Adapted the method to clarify we have calculated the weighted average
- Page 7 Line 191-194, Page 8 Line 218-222: Results - Changed baseline fat fraction and change in fat fraction values in the text to weighted average
- Page 10 Line 267-291: Results - We have now also applied this weighting when averaging the 5 separate slices and updated the values in the text where needed.
- Page 30/31 Figure 7: Replaced the figures with the new values applying weighted averaging to the 5 slices
- Page 33/34 Table 1: Changed baseline fat fraction and change in fat fraction values in the table.

9. P5. What anatomical landmarks were chosen to determine the muscle extremities? How was the muscle length taken into account to determine and compare the FFs between individuals with different limb lengths? In other words, how was the segment location determined? As a percentage of the muscle lengths? or in cm? from which anatomical landmark?

Response: Data-acquisition - The most proximal slice of the proximal image stack was positioned at the head of the femur. Subsequent stacks were placed distal with 45 mm overlap between stacks, until the lower legs covered from hip to ankle. For most participants, 3 stacks were sufficient, but for one tall participant a 4th stack was needed.

Data-analysis: Muscle extremities were defined as the last distal and the last proximal slice where the muscle was still large enough to reliable segment, *e.g.* constituting of at least several voxels. In all individuals, this approach resulted in a full muscle segmentation from the distal to proximal for most muscles. There are a few exceptions, in some patients the imaging slices did not cover the most distal (EDL) or proximal muscle extremity (VL, RF, SAR, GRA, AM, and AL). However this was never more than ~2 cm (~5%) of the muscle length and is therefore not expected to affect our results.

The approach described above ensured the same anatomical coverage between individuals during both data-acquisition and segmentation. However, this does mean that the number of segmented slices differs between muscles and individuals. To compare FF's between individuals and muscles, we expressed each slice number as the percentage of the muscle length. For each muscle, the first segmented distal slice was set at 0%, and the most proximal segmented slice was set at 100%. All other slices were equally divided between 0 and 100%. This approach explains the x-axis shown in figure 2A and figure 4.

For the quantitative comparison, we divided each muscle up in five equally spaced segments, based on the percentage of the total muscle length, 0-20% (distal 1/5th of the muscle), 20-40%, 40-60%, 60-80%, and 80-100% (proximal 1/5th of the muscle).

We have clarified the above in the manuscript text:

“Axial 3D 2pt Dixon images were acquired to quantify muscle fat infiltration. The lower extremity was covered from hip to ankle using 3 or 4 stacks. The most proximal stack was placed with the proximal slice at the head of the femur and each subsequent stack was placed distal to the former stack with 45 mm overlap” Page 4 Line 94-97

“To evaluate the fat infiltration along the proximal-distal axis, we also calculated the fat fraction per slice for each muscle. Because muscle length and therefore the number of segmented slices differed between muscles and participants, slice numbers were expressed as percentage of the muscle length with the most distal slice set at 0% and the most proximal slice set at 100%. Furthermore, we divided each muscle into five equally spaced proximal-distal segments, again relative to the muscle length, and calculated the average fat fraction per segment. For example, the most distal 1/5th of the muscle was set at 0-20% and the most proximal 1/5th of the muscle set at 80-100% of its length.” Page 5 Line 135-141

10. P6 l142 and followings : « To estimate the size of this effect in FSHD, we compared the baseline fat fraction and change in fat fraction over the five middle slices to the whole muscle analysis using a Bland–Altman analysis. Furthermore, we calculated the absolute difference in baseline fat fraction and change in fat fraction over the middle five slices compared with the whole muscle ». Knowing the uneven distribution of FF along the muscle length, it would have been

fair to also compare measures performed on only 5 slices but more separated on the muscle length.

Response: We agree with the reviewer that this slice repositioning is an interesting alternative and have therefore now incorporated this in the paper. We have estimated the baseline fat fraction and change in fat fraction for five slices equally spread over the whole muscle length by selecting for each muscle slices at 10%, 30%, 50%, 70%, and 90% of the muscle length. The slice positions corresponding to these percentage of the muscle length were calculated as explained in our response to point 9 (see above).

The results indicate that fat fraction determination of five slices evenly distributed over the length of a muscle provides a good approximation of the whole muscle fat fraction. The estimated absolute error in the baseline fat fraction drops from 3.7% for the five slices in the middle of the muscle to 0.6% for the five slices spread along the muscle. The same holds true for the change in fat fraction, this absolute error decreases from 0.6%/year to 0.2%/year. For both baseline fat fraction and change in fat fraction, the errors are now independent of baseline fat fraction in the whole muscle.

We have added this data into the paper:

- Method section: Page 6 Line 167-170
- Results section: Page 10 Line 267-291
- Discussion section: Page 12 Line 333-335 and Page 14 Line 429-431 and Page 15 Line 435-436
- We added additional figures to Figure 7. Page 30/31

11. P7l173 : « The presence of a fat gradient with the highest fat fraction distally was further demonstrated by quantitative assessment of the five muscle segments (Fig 2C–F). » I did not read how the segments were defined. Please clarify.

Response: Please see response to point 9 and the clarifying text in the manuscript Page 5 Line 135-141

12. P8l218: what was the length covered by the 5 adjacent slices?

Response: The 5 adjacent slices cover a total length of 2.5 cm (5 mm per slice with no slice gap). We have added this 2.5 cm to the text (Page 10 Line 271)

13. Figure 2 : how was the position along the leg calculated in %? Was the slice thickness (stack length) adjusted to the leg length. If the slice thickness was fixed, how were the FF transformed in FF in % along the leg muscle. Also, as already asked, what were the anatomical landmarks chosen for 0 and 100%.

Response: Please see our response to point 9. The FF were transformed in FF in % along the leg muscle for each muscle individually, with 0% the most distal segmented slice for that particular and 100% the most proximal segmented slice for that particular muscle. We have tried to clarify this better in the legend of figure 2.

“Fat fraction displayed at each position along the gastrocnemius medialis expressed as a percentage of muscle length from the most distal segmented slice (0%) to the most proximal segmented slice (100%)” (Page 24 Line 623-624)

14. Figure 4: nice examples of the front progression of fatty replacement but in Figure 5 : no evidence of such front progression at fatty replacement.

Response: Please see our response to point 1.

Response to Reviewer #3:

The article “Whole-muscle fat analysis identifies distal muscle end as disease initiation site in facioscapulohumeral muscular dystrophy” is a solid contribution to the muscle MRI literature on FSHD. Its main focus is on seven male patients and 298 of their leg muscles longitudinally over 3.5 years. The main novel findings are that 1) distal muscle are the most affected and 2) how we sample muscle slices affects our summation of fat fraction as the fat is distributed unequally across the muscle.

This brings into attention that our prior ways of averaging only the central slices subjects our analyses to more variability; and is helpful to future analyses that we should calculate more slices.

The methods for acquisition and analysis of the muscle MRI were sound. The results and graphs are well presented, specifically Figures 2-5 are especially helpful to push forward the field. The discussion was reasonable and well written. Some minor points:

1. Describe more the muscles that were exception to the rule of the distal-to-proximal gradient. 43% had clear decreasing fat fraction; 34% of muscle (low fat infiltration) and 11% of all muscles (high fat infiltration) had no or minimally decreasing fat fraction. Leaving 12%/36 muscles that did not follow the rule. An appendix table of those muscles, in terms of their position and amount of fatty infiltration at baseline, or more examples of those muscles not showing the gradient would be helpful (a full figure 6 of the other muscles).

Response: Thank you for this valuable suggestion. We have created supplemental Table 1 showing all muscles that do not fulfil the prevailing distal-to-proximal fat infiltration pattern including their baseline fat fraction levels. We have slightly modified the analysis and focussed it on muscles with a mild-to-moderate fat fraction, i.e. between 10% to 60% fat. The reason being that the pattern is difficult to judge in the very highly fat infiltrated muscles and this way we are more consistent with the rest of our analysis. Of the 170 intermediate fat infiltrated muscles, 56 were in first instance an exception to the rule. However, a closer look revealed that 28 of them had a homogenous fat fraction level that is only just elevated and might also be considered normal. Therefore, we have split our table into two groups; supplemental table 1A describes these former muscles, while supplemental table 1B describes the muscles that show clearly a different pattern. We have updated the text (Page 9 Line 244-265) and Figure 6 (Page 29) accordingly.

2. Did the authors see small areas of T1 fat in the muscles that coalesced as the fat progressed from distal to proximal area. I.e., while there is a distal -> proximal march of fat, did the authors see small pockets of T1+ fatty infiltrated areas that would be made confluent over time.

Response: We did not acquire any T1 weighted images, but only Dixon images. Based on visual assessment of these Dixon scans we can say that there were no small pockets visible along the muscle that became confluent over time.

The authors measured the fat fraction on every fifth slice and then coalesced the fat fraction into 5 equally distributed segment along the muscle axis. Did the authors go back and check whether the coalesced 5 segments in the baseline and follow-up MRIs corresponded to roughly the same areas, or was there also noise and inexactness in determining the 5 segments? If that occurred, does that change the findings.

Response: We acquired our data without a slice gap. Furthermore, during the post-processing, we aligned the baseline and follow-up scans by eye and ensured that the first and last slice matched between baseline scan and follow-up scan. Subsequently every slice was segmented. Consequently, the five proximo-distal segments and the anatomical muscle segmentation in those scans will also align and correspond to almost exactly the same area for baseline and follow-up. We therefore do not believe that inaccurate matching and baseline and follow-up scans has affected our findings.

3. The description of Figure 7D was slightly confusing to the reviewer and would be benefited by describing it better/more in the legend and the results section.

Response: We have changed the legend of figure 7B and hope it is now clear for the reviewer. Furthermore, we have changed the legend label in the figure to error instead of difference so it matches with the legend text.

“D) Tukey boxplots (showing median, interquartile range, and outliers) of the error (white) and absolute error (grey) for individual muscles, with the muscles grouped based on whole-muscle baseline fat fraction (10% increments). The grey bars shows that the change in fat fraction is generally overestimated in muscles with a baseline fat fraction of 20–40% and underestimated in muscles with a baseline fat fraction of 40–60%. The white bars shows that in intermediately fat infiltrated muscles (10-60% baseline fat fraction), the absolute error was about 0.8%/year.” (Page 31 Line 690-707)

4. Expound more in the discussion on the limitation that this study also only samples every 5th slice (due to the time intensity of segmenting and analyzing each slice) and that ideally, sampling as many slices from the distal to proximal end would probably provide the most accurate description of the muscle fat burden. This does not affect the paper’s central posit that a distal-to-proximal gradient is present in most muscles, could affect the summed average fat fraction.

Response: We like to clarify that in this study we sampled every slice, instead of every 5th slice. The reviewer is correct in that we manually segmented muscles in every 5th slice (slice 1, 6, 11 etc), however, after that we applied an algorithm to automatically segment the remaining slices (slice 2, 3, 4, 5, 7, 8, 9, 10, etc.). See also figure 1. Since the data was acquired without a slice gap, this means that muscle was segmented every 5 mm. The average fat fractions presented per muscle, muscle group or whole lower extremity are to our knowledge therefore as accurate as possible.

We have emphasized this in the legend of figure 1: “As a result, all muscles were segmented on every slice (every 5 mm).” (Page 23 Line 615-616)

REVIEWERS' COMMENTS:

Reviewer #1 (Remarks to the Author):

The authors have addressed all my comments appropriately.

Reviewer #2 (Remarks to the Author):

I am happy with this revised version.

The authors have addressed the questions and comments that I and the other reviewers had raised.

Reviewer #3 (Remarks to the Author):

I want to thank the authors for their detailed revisions and rebuttal. Additional data was released with the TIRM data added. Furthermore, I appreciated the additional analysis on non-standard progressing muscles. A few notes:

1. I disagree with the characterization of TIRM positivity as “edematous lesions”. To my knowledge there is no histopathological correlation of muscle edema for STIR/TIRM positivity.
2. I still having problems understanding the paragraph on “The effect of localised sampling on estimated change in fat fraction.” I am not sure “error” is the best word, and I believe “error” is the variance from the fat fraction derived from the central five slices to the whole muscle and “absolute error” is the variance from dispersed five slices to the whole muscle? I am not sure what is absolute about that characterization.

RESPONSE TO REVIEWER'S COMMENTS

We thank the reviewers and are glad to hear they are happy with our revision of the manuscript with the title:

"Whole-muscle fat analysis identifies distal muscle end as disease initiation site in facioscapulohumeral muscular dystrophy" by L. Heskamp, A. Ogier, D. Bendahan, A. Heerschap. (COMMSMED-22-0029-A)

Our response to the remaining comments of reviewer 3 is indicated in blue below. Changes in the manuscript in response to the comments of reviewer 3 are labeled via comments and highlighted via tracked changes.

Response to Reviewer #3:

I want to thank the authors for their detailed revisions and rebuttal. Additional data was released with the TIRM data added. Furthermore, I appreciated the additional analysis on non-standard progressing muscles. A few notes:

1. I disagree with the characterization of TIRM positivity as "edematous lesions". To my knowledge there is no histopathological correlation of muscle edema for STIR/TIRM positivity.

We thank the reviewer for this comment. Although it is widely assumed that STIR/TIRM positivity is a sign of oedema, indeed no histopathological proof has been reported of a correlation between STIR/TIRM positivity and oedema in FSHD. Instead a correlation has been observed between STIR/TIRM positivity and inflammatory markers like CD8+ (Frisullo et al. 2010, <https://doi.org/10.1007/s10875-010-9474-6>). Therefore we have replaced the term edematous by inflammatory in the abstract (Page 3, Line 39 and Line 48), and in the methods section Page 6, Line 129.

2. I still having problems understanding the paragraph on "The effect of localised sampling on estimated change in fat fraction." I am not sure "error" is the best word, and I believe "error" is the variance from the fat fraction derived from the central five slices to the whole muscle and "absolute error" is the variance from dispersed five slices to the whole muscle? I am not sure what is absolute about that characterization.

We understand that this paragraph may be confusing. Therefore we now describe in the Methods section under Statistics what our definition of error and absolute error is.

With error, we meant the difference between the baseline fat fraction obtained over one of the five slice approaches and the baseline fat fraction obtained when analysing all slices. And In other words, error = fat fraction over 5 slices – fat fraction over the whole muscle. With absolute error, we meant the absolute value of this difference. Similarly, error and absolute error are calculated for change in fat fraction.

Hence, error and absolute error are both determined for the central five slices approach and for the dispersed five slices approach and are calculated for each individual muscle. The Bland-Altman analysis reports the average value of the errors (also known as the bias) over all muscles and the linear mixed model (to test if the absolute errors deviate from 0) reports the estimated absolute error over all muscles.

We have now tried to clarify it better in the manuscript text:

- 1) We have rewritten the statistics paragraph to better define “error” in the Methods section (Page 8-9 Line 190-203) and Results section (Page 12 Line 296-299)
- 2) We explain that when we report the statistical test of the absolute error this is estimated by a linear mixed model (estimated absolute error). Page 9 Line 202-203 and Page 12 297-299
- 3) We use the wording average error when reporting the results of the Bland-Altman analysis. We refrained from using bias to avoid confusion. Page 9 Line 199-200 and Page 12 300-319.
- 4) Removed any inconsistent use of the wording bias, error and difference in the Results and Discussion sections. Page 12 Line 290-319, and Page 16 Line 443.

Furthermore, we have updated figure 7 and emphasize with text (titles) above the figures more clearly that the right side graphs represent data for the central five slices vs. the whole muscle and the left side graphs represent data for the dispersed five slices vs. whole muscle.